# Head–Neck Cancer Delineation

Enrico Antonio Lo Faso †, Orazio Gambino *,† and Roberto Pirrone

Department of Engineering (DI), University of Palermo, Viale delle Scienze, Ed.8, 90133 Palermo, Italy;
enricoantonio.lofaso@community.unipa.it (E.A.L.F.); roberto.pirrone@unipa.it (R.P.)
* Correspondence: orazio.gambino@unipa.it
† These authors contributed equally to this work.

**Abstract:** Head–Neck Cancer (HNC) has a relevant impact on the oncology patient population and for this reason, the present review is dedicated to this type of neoplastic disease. In particular, a collection of methods aimed at tumor delineation is presented, because this is a fundamental task to perform efficient radiotherapy. Such a segmentation task is often performed on uni-modal data (usually Positron Emission Tomography (PET)) even though multi-modal images are preferred (PET-Computerized Tomography (CT)/PET-Magnetic Resonance (MR)). Datasets can be private or freely provided by online repositories on the web. The adopted techniques can belong to the well-known image processing/computer-vision algorithms or the newest deep learning/artificial intelligence approaches. All these aspects are analyzed in the present review and comparison among various approaches is performed. From the present review, the authors draw the conclusion that despite the encouraging results of computerized approaches, their performance is far from handmade tumor delineation result.

**Keywords:** head–neck cancer (HNC); head and neck squamous cell carcinoma (HNSCC); nasopharyngeal cancer (NPC); segmentation; tumor delineation; CT; PET; MRI





## 1. Introduction

Head and Neck Squamous Cell Carcinoma (HNSCC) or Head and Neck Carcinoma (HNC) is one of the most common malignancies by incidence worldwide and includes cancers of the upper aerodigestive tract (oral cavity, oropharynx, hypopharynx and larynx and so on) [1–3]. The onset of the disease could be due to various factors: hpv infection [4], genetic inheritance, ingestion or inhalation of harmful substances both voluntarily (tobacco and alcohol [5]) and involuntarily, in case of exposure to toxic substances dispersed in the environment [6]. Patients affected by inoperable HNC must be treated with radiotherapy (RT), so the delineation of tumors and metastatic limph nodes is a fundamental task of RT planning and it must be performed on radiological images, usually Positron Emission Tomography (PET) and Computerized Tomography (CT) scans. PET scan is a non-invasive radiological examination regarding functional imaging, which reveals metabolic changes of the tissues providing in vivo important measurements about cancer's biological evolution. Such an examination can be performed only after administering a radiotracer to the patient. A typical PET radiotracer used in the evaluation process of HNC is the glucose analog 18F-fluoro-2-deoxy-D-glucose (FDG), which is a weak radioactive substance. High metabolic rate tissue, like cancer, increases its FDG uptake, which is revealed by detectors. The detection of a high concentration of FDG reveals primary cancers and metastases appearing as "hot spots" surrounded by "cold" non-pathological tissue in a PET image. As a consequence, PET images are preferred to CT scans for their high-contrast between cancers and the rest of the tissues, but they exhibit a low spatial resolution. CT scan is a common medical imaging procedure combining a number of X-ray measurements performed at different angles. The resulting image is a cross-sectional view (slice) of a volume revealing the morphology of the internal organs: a bright pixel represents a high-density volume element, whereas a dark pixel is associated with a low-density one, so an

appropriate contrast medium must be used to visualize particular parts of the human body. Despite a high spatial resolution, the contrast of such images is low for the HNC. For this reason, often the methods make use of multi-modal data (usually PET-CT) to compensate for the respective lack of image features (spatial resolution for PET, low contrast for CT). Often these images are merged to create a new image that is used for tumor delineation. Recently, Magnetic Resonance (MR) scans have been considered for the HNC. MR Imaging (MRI) makes use of an intense magnetic field and electromagnetic waves to obtain detailed images of the organs and tissues of the human body. Like a CT scan, it provides a cross-sectional view of the human body and can produce images with different contrast among the tissues in the function of the type of electromagnetic waves (T1-weighted, T2-weighted and so on). Even if it exhibits a good spatial resolution and the contrast among tissues is better than CT, it is not sufficient to distinguish the neoplastic tissue from the surrounding one, so also in this case MR is used in combination with PET [7,8]. The usefulness of using PET and MRI combined together in the evaluation of head and neck cancers is also investigated in the study of Kogaczewska et al. [9]. In the end, the delineation of tumors is a task making use of multi-modality imaging but, when it is manually performed, it incurs several problems: it is time-consuming, labour-intensive and prone to inter- and intra-observer variations [10,11]. In Figure 1 the comparison among PET, CT and MR images are shown. As a consequence, the manual inconsistency affects the result of a hand-made segmentation and for the same reason the gold standard is affected by the same issues, because it is often obtained by manual segmentation. Finding an automatic/semi-automatic, robust and precise method to delineate the neoplastic formation is an important field of research to overcome the issues mentioned above for the manual segmentation. The present review collects papers regarding HNC considering all the diagnostic modalities and methods adopted to perform an automatic and semi-automatic segmentation. The paper is organized as follows: Section 2 describes the methodology adopted to collect the papers, Section 3 describes the deep learning methods, Section 4 reports all the other methods, Section 5 is dedicated to the comparison between these two approaches, Section 6 reports some conclusions.

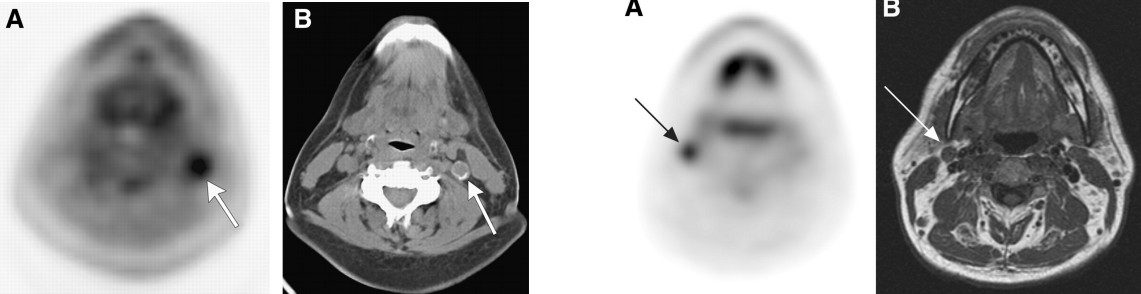

**Figure 1.** Head–Neck cancer, transverse plane. The arrows indicate the lesions. Left: (**A**) Positron Emission Tomography (PET) (**B**) Computerized Tomography (CT) [12]. Right: (**A**) PET (**B**) T1-weighted Magnetic Resonance Imaging (MRI) [13].

## 2. Methodology

This review contains papers that have been selected by using the search engines of the main publishers in the scientific literature: MDPI, Association for Computing Machinery (ACM) Digital Libraries, Institute of Electrical and Electronics Engineers (IEEE) Xplore, Springer and Elsevier. The search has been extended by consulting the main public search engines, such as PubMed and Google Scholar. The query has been subdivided into three groups: modality ("PET", "CT", "MRI"), main theme ("head and neck", "head–neck", "HNC", "HNSCC") and operation ("segmentation", "delineation"). All the permutations among these three groups generate a variety of queries. Groups "main theme" and "operation" aren't sufficient to isolate only biomedical image processing papers, because other topics like molecular research and therapy can be found. The group "modality" helps to find useful paper in a precise way. In order to avoid papers regarding trivial approaches,

only recent peer-reviewed papers have been taken into consideration and belonging to Journal with a relevant interest in this field of research. Both duplicate and non-pertinent items have been removed, after screened title, keywords and abstract of each paper. Due to arising deep learning methods developed in recent works, the analysis distinguishes between this family of approaches and the others.

## 3. Deep Learning Methods

In this section, a list of methods based on deep learning techniques are presented. They are ordered for image modality, starting from MR images and finishing with the PET/CT multimodality. Most of them perform automatic segmentation with the use of convolutional neural networks (CNN). For this reason, before proceeding with the description of the methods, some concepts recurring in these methods are introduced.

### 3.1. Convolutional Neural Network CNN

Convolutional Neural Networks are a specific case of feed-forward neural networks. They are made of neurons with learnable weights and biases, just like classic neural networks. The CNNs take images as input, assigning importance to various aspects in the image and differentiating one from the other. A CNN can successfully capture the spatial and temporal dependencies in an image by applying relevant filters which are adjusted during the training phase to understand the features of the image better. The Convolution Operation aims to extract the high-level features from the input image, the low-level features are generally extracted in the first layers, whereas the last layers extract higher level features.

### 3.2. Pooling Layer

Another important concept is the pooling layer, which produces a summary statistic of its input in order to reduce the spatial dimension of the feature map. The max-pooling reports the maximal values in each rectangular neighborhood of each point (i,j), the average pooling reports the average values.

### 3.3. Optimizers

All the methods based on deep learning use an optimization algorithm. It changes the neural network parameters in order to reduce the loss function and providing the most accurate results possible. The most used optimizers are the Stochastic Gradient Descent (SGD) and the Adaptive Moment Estimation (Adam) [14]. The SGD algorithm is an approximation of Gradient Descent (GD), since it substitutes the exact value of the gradient in the cost function with an estimated value obtained by evaluating the gradient only on a subset of the addends. For this reason, it is less computationally expensive than the GD algorithm. The Adam optimizer is an extension of the SGD that has recently seen broader adoption for deep learning applications. While SGD, in the standard implementation, maintains a single learning rate for all weight updates and it does not change during the training, Adam computes different learning rates for different parameters, it uses estimations of the first and second moments of the gradient to adapt the learning rate for each weight of the neural network. The n-th momentum of a random variable is defined as the expected value of that variable to the power of n. Specifically, Adam computes an exponential moving average of the gradient and the squared gradient, and the parameters $\beta_1$ and $\beta_2$ control the decay rates of these moving averages.

### 3.4. Dice Similarity Coefficient (DSC)

In order to evaluate the accuracy of each method, they are compared using the same metric, which is the Dice Similarity Coefficient (DSC). In particular, this coefficient is

applied to Boolean data, using the definition of true positive (TP), false positive (FP) and false negative (FN) and it can be written as

$$DSC = \frac{2TP}{2TP + FP + FN}$$

It represents the Harmonic Mean of Precision and Recall. It means that the DSC directly depends on both precision and recall. The DSC penalizes models in which the accuracy hangs mainly on one of the metrics between precision and recall. For this reason, DSC is generally the most used metric for describing the performance of a model.

*3.5. Methods Review*

In B. Zhao et al. [15] is presented an end-to-end algorithm for the segmentation of head and neck tumors from MRI images. A total of 163 MRI images (2D slices) from 17 patients from the Beatson west Scotland cancer center are used in this experiment. The labels are manually delineated by clinicians from Beatson. This dataset is split into three subsets, the first two are used for training and the third for the test. The proposed algorithm is a neural network similar to U-net [16], with the addition of a multi-scale feature extraction process. While to access a larger receptive field there are some drawbacks like many layers or bigger kernel sizes, thanks to the dilated convolution, it is possible to look at a larger context without occurring in the reduction of feature resolution. The extracted multi-scale features are combined using concatenations to improve the performance of the network. The dataset is augmented by rotation, zoom, shift and flip in order to improve generalization, Batch normalization is also used to improve the learning process optimized using Adam optimizer. The average DSC of the cross-validation is 0.644 which is about 0.05 higher than the original U-net.

Lars Bielak et al. [17] analyzes the contribution of 7 MRI input channels in a CNN for the segmentation of Head Neck cancer. The dataset of this work contains a total of 33 MRI of patients with head-neck cancer (HNC), the Gross Tumor Volume (GTV) delineation is performed by expert radiation oncologists and radiologists mainly on T1- and T2-weighted images. All the images are co-registered and interpolated to a common base resolution of $0.45 \times 0.45 \times 2$ mm$^3$ and normalized to a standard deviation and mean of 0.25 each. The described CNN is based on the DeepMedic architecture [18], with the use of two pathways, one with the original resolution and the other subsampled of a factor of 3. The network is composed of 10 convolutional layers each with 104 feature maps and residual connections in layers 4, 6, 8 and 10. A 20% dropout is set in each layer and the Dice loss function is used to train the network. This system uses a patch-based approach, the patch size is $38 \times 38 \times 8$ and $78 \times 78 \times 8$ pixels respectively for each path. The CNN is trained with all seven input channels and also in other seven configurations in which one of the input channels is left out. Trying all the different configurations allows understanding which of the seven inputs carries a greater impact in this task. The final segmentation performance is evaluated with the DSC whose value reaches 0.65.

Another study made by Yufeng Ye et al. [19], proposes an automated segmentation method for nasopharyngeal carcinoma based on convolutional neural network (CNN) on a dataset containing 44 T1- and T2-weighted MRI images for a total of 1950 pairs of image slices. A couple of images of each patient are co-registered in order to make use of the information of both the modalities. The images are resampled by using linear interpolation, min–max normalization is performed and all the slides of the images are zero padded and cropped to $256 \times 256$ pixels. The network is divided into an encoder part and a decoder part: the encoder extracts the features of the images thanks to four encoder blocks and a dense connectivity block, the decoder uses deconvolutions to recover the extracted features to the initial input size. The encoder is composed of three $3 \times 3$ convolutional layers, two group normalization layers [20] followed by leaky rectified linear unit (LReLU) layers [21]. The decoder consists of five decoder blocks, each block is composed of a $3 \times 3$ deconvolutional layer, a concatenation layer and two $3 \times 3$ convolutional layers followed

by two LReLU and GN layers. Thanks to the concatenation layer, the feature maps are fused to address the problem of losing information after the deconvolutions. The final layer is a $1 \times 1$ convolutional layer with a sigmoid which determines the final tumor segmentation. Dice loss is used to optimize the training process. The performance of the method was evaluated using a 10-fold cross-validation strategy and three different input cases: the T1 only which obtained a DSC of 0.62, the T2 only with a DSC of 0.64, the combination T1+T2 with a DSC of 0.72.

The method of Zongqing Ma et al. [22] proposes an automatic segmentation algorithm for 3D T1-weighted MRI, it is developed relying on a dataset of 30 patients with different cancer stages. The resolution of all the images is $528 \times 528 \times 290$ and the voxel size is $0.61 \times 0.61 \times 0.8$ mm$^3$. Intensity normalization is performed by using the method proposed by Nyúl et al. [23] and then an isotropic resampling was applied to obtain a resolution of $1.0 \times 1.0 \times 1.0$ mm$^3$. For the segmentation task, a CNN architecture similar to Alex-Net [24] is used, it takes image patches as input and it is composed of 5 convolutional layers and three fully connected layers. After the first, second and last layers a max-pooling layer is used to make the network learn space-invariant features. In the end, the final binary segmentation is performed by a Softmax layer. A Dropout of 0.5 is also used in the first two fully-connected layers in order to avoid overfitting. The 3D segmentation is performed in three different paths which correspond to each orthogonal perspective (axial, sagittal or coronal), then the voxel probabilities are computed averaging the probabilities of the three paths. In order to refine the segmentation, a graph cut algorithm is used to interpret the image as a graph and to solve an energy minimization problem. The network is trained for 10 epochs using 0.001 learning rate, 0.0005 weight decay, 0.9 momentum and a batch size of 100. The method is evaluated by using leave-one-subject-out cross-validation obtaining a DSC value 0.851.

Finally, Qiaoliang Li et al. [25] developed the deep learning method which obtains the best DSC score on MRI images. In this article is presented a CNN architecture based on a dataset of 29 patients from the First Affiliated Hospital, Sun Yat-Sen University and ground truth manually delineated by two experienced radiologists. A total of 87 slices of contrast-enhanced MRI (CE-MRI) is extracted, containing only the tumor area of each patient. These images are augmented to more than 60,000 slices performing rotation, changing contrast and adding random Gaussian noise. Then the images were normalized performing z-score normalization. The feature extraction phase consisted of 4 Conv-ReLu blocks and 2 Pool-Conv-Relu blocks which transform the $144 \times 144$ input images into $36 \times 36$ feature maps, the reconstruction phase consisted of two deconvolutional layers to obtain an output image with the original size of $144 \times 144$ pixels. The network is trained with a learning rate of $10^{-7}$, step size of $10^5$, momentum of 0.9 and weight decay $5 \times 10^{-4}$. The slices of 28 patients are used for training while the slices of the remaining patient were used for testing. The precision of the method measured in terms of average DSC is 0.89.

In Sahar Yousefi et al. [26] method based on CT images known as DenseUnet is described. It is a 3D network composed of a contractile path to extract the image features and an expanding path to recover the original input patch resolution. Each path is composed of dense blocks, down-sampling units and up-sampling units. Skip connections are used to assist the network to retrieve the lost information after the down-sampling process. The proposed architecture is trained and tested on a dataset containing 553 esophagus CT images from 49 distinct patients. In order to reduce memory consumption, this approach uses 3D patches rather than complete scans. The main component of the network is the dense block, it contains one conv($1 \times 1 \times 1$)-BN-ReLU and one conv($3 \times 3 \times 3$)-BN-ReLU. The down-sampling block is a conv($1 \times 1 \times 1$)-BN-ReLU-MaxPool and the up-sampling conv($3 \times 3 \times 3$)-BN-ReLU-deconv($3 \times 3 \times 3$). The final part of the network consists in another convolutional layer and a soft-max layer in order to compute the output which can be classified as Gross Tumor Volume (GTV) or background. The network is trained for 10k iterations with a batch size of 20, for the optimization is used the Adam optimizer with a learning rate of $10^{-4}$. The dataset containing 553 scans from 49 distinct patients is

split into 30 patients for training, 6 for validation and 13 for testing. During every iteration, data augmentation is performed adding withe noise to the input patches. Different tests are done varying the number of sub-blocks and also the number of feature maps inside each sub-block and are computed the DSC for each configuration to compare them and to find which one is better. The best configuration of the network was DenseUnet$_{122}$ which achieves a DSC value of 0.73.

Remaining within the scope of CT images Men et al. [27] develops a Deep Deconvolutional Neural Network (DDNN) consisting of two components: an encoder and a decoder network. The encoder is made of 13 convolutional layers and it is intended to extract feature maps of the input images, then the decoder will recover the original resolution of the images by deploying deconvolution. The encoder layers are based on the VGG-16 architecture [28], which is well-known for feature extraction. In addition, for this task, the fully connected layers were replaced with fully convolutional layers. The dataset used in this work consists of 230 CT images of patients diagnosed with Nasopharyngeal Cancer (NPC). Radiation oncologists contoured the nasopharynx gross tumor volume (GTVnx), the metastatic lymph node gross tumor volume (GTVnd), clinical target volume (CTV), and organs at risk (OARs) in the planning CT. Images from 184 patients were randomly chosen to be part of the training set and the remaining six patients were used to evaluate the performance of the model. The initial learning rate was set to 0.0001, learning rate decay factor to 0.0005 and decay step size to 2000. The network is trained until the precision on the training set converged and then it was tested on the validation set. The DSC values how this method outperformed the VGG-16, obtaining an average score of 75.3% $\pm$ 11.3% compared to 59.9% $\pm$ 22.7% of the VGG-16.

The work of Bilel Daoud et al. [29] proposes a deep-learning method for the nasopharyngeal carcinoma (NPC) segmentation from CT images. The dataset consists of 70 CT images of patients with NPC, for each patient all the contours of cancer are traced from two radiation oncologists. The method includes two phases: during the first phase, the non-target organ regions are eliminated from CT images, during the second phase the NPC is detected from the remaining part of the images after the first step. The system is composed of three different paths for each section (axial,coronal and sagittal) for a total of six CNNs: three for the detection of non-target organ region and three for the detection of NPC. After the whole process, the output of the second path is integrated into a single image containing the segmented tumor. The CNNs are composed of three convolutional layers with max-pooling and ReLu activation function, and one fully connected layer. All the convolutional layers use filters with a size of 3 $\times$ 3 and a stride of 1 with the same padding. We developed two different systems, the first uses fixed size patch size and the second variable patch size as input images. To evaluate the applicability of the proposed systems, the dataset is divided into seven sub-datasets, each of which contains 10 patients. Six sub-datasets are used in the training phase while the last one is used to test the performance of the trained network. The performance is computed in terms of DSC that reaches 0.87 for the variable patch size approach and 0.83 for the fixed one.

Other methods have been developed using combinations of different image modalities. Vincent Andrearczyk et al. [30] used NiftyNet [31] in order to implement V-net, a fully-convolutional neural network in 2D and 3D versions. The dataset is composed of PET/CT images of 202 patients and the relatives Gross Tumor Volumes (GTVs) manually delineated by professional radiation oncologists. This dataset is extracted from the one proposed in (Valli'eres et al. 2017) [32], and also avaible on The Cancer Imaging Archive (TCIA) in the context of radionics studies, but focusing only on 202 oropharynx tumors. The authors resampled the PET and CT volumes to an isotropic 1 $\times$ 1 $\times$ 1 mm voxel spacing using trilinear interpolation. Then they cropped the images to a volume of size 144 $\times$ 144 $\times$ 144 voxels which is centered in the oropharyngeal region, excluding other regions of little importance for this task. The architecture uses four downsampling blocks and four upsampling blocks and the final prediction is made by a residual convolutional block. The functions of downsampling and upsampling are performed by convolutional layers with 2 $\times$ 2 $\times$ 2 filters

using a stride of 2. In total, there are 30 convolutional layers with ReLU activations and final softmax activation. Both 2D and 3D versions of the network are trained using an Adam optimizer with a batch size of 12, a learning rate of 0.0003 for 200 iterations. In order to evaluate the performance of these models, a leave-one-center-out cross-validation was performed. This study showed that the two modalities are complementary and there is a statistically significant improvement from 48.7% and 58.2% Dice Similarity Coefficients (DSC) with CT-only and PET-only segmentation respectively, to 60.6% with a bi-modal late fusion approach. This study shows that the 2D approach slightly outperforms the 3D one on this specific task. (60.6% vs. 59.7% respectively).

The complementarity between PET and CT images is also analyzed in Zhe Guo et al. [33]. this article describes a Dense-Net framework, an automatic segmentation CNN based on 3D convolution with dense connections to enable better propagation and take full advantages of the features extracted at each level from multi-modality images. This network is trained and evaluated on the dataset (Martin et al. 2017) [32] of HNC from the Cancer Imaging Archive (TCIA) (Clark et al. 2013) [34], focusing on 250 PET/CT images of patients with Head-Neck cancer. The dataset is split into training, validation and testing, respectively into 140, 35 and 75 patientes. The images are cropped to a 3D volume with $128 \times 128 \times 48$ pixels in order to reduce memory consumption. The network structure is based on Jègou et al. [35] with dense blocks and transition-down and transition-up modules. The whole dense block is constructed using four convolution layers, the feature maps from all four layers are concatenated constituting the output of the dense block. This architecture contains nine dense blocks, four transition-down and four transition-up modules. Dense connection introduces an extreme connecting pattern, it links a layer to all its subsequent layers using skip connection, extremizing the concept of residual connections [36]. The quantitative accuracy of this method, measured in terms of DSC is 0.73 for the multi-modality approach, while for the PET single modality is only 0.67.

Yngve Mardal Moe et al. [37] develop a method based on the U-Net [16] architecture but using PET and CT images. The Gross Tumor Volumes (GTVs) are delineated by professional oncologists from 197 Head Neck Cancer (HNC) patients planned for radiotherapy at Oslo University Hospital between January 2007 and December 2013. The dataset contains 197 Head Neck Cancer (HNC) patients planned for radiotherapy at Oslo University Hospital between January 2007 and December 2013. It is split into three parts: 142 patients for the training phase, 15 for the validation phase and 40 patients for testing. Then PET and CT images are co-registered and the model is trained for both single (CT-only and PET-only) and bi-modal (PET/CT) approaches. In order to train the U-Net Adam optimizer is used with a learning rate of $10^{-4}$ and standard parameters. After each activation function, batch normalization is used to make the training process more stable. The method is evaluated with both Cross-Entropy and Dice loss functions. The validation set is useful to explore the hyperparameters configuration that provided the best precision in terms of the Dice similarity coefficient. Then, the generalization ability of the algorithm is tested on the 40 patients of the test set. The results in terms of Dice coefficient are the following: CT: $0.65 \pm 0.17$, PET: $0.71 \pm 0.12$, PET/CT: $0.75 \pm 0.12$

Dakai Jin et al. [38] aim to exploit the complementary information within PET and CT imaging spaces. For this purpose, a two-stream chained 3D deep network fusion pipeline is designed. This network uses early and late 3D deep network fusions of CT and PET images. The images are co-registered applying the cubic B-spline and using the lung mass centers of the CT scans as initial matching position. The mass center is produced by the P-HNN model, which generates a segmentation of the lung field even in severely pathological cases, generating aligned PET/CT images. First, two separate streams generate segmentation maps using the CT images and the early fusion PET/CT. The functioning of the pipeline can be seen as a late fusion of the CT and early fusion models. To effect the fusion and segmentation, a progressive semantically nested network (PSNN) is proposed. It consists of a set of $1 \times 1 \times 1$ 3D convolutional layers which collapse the feature maps after each convolutional block into a logit image. Then, this image is combined with the

image previously generated by the higher level segmentation and finally, an aggregated segmentation map is created. The PSNN is trained using four deeply-supervised auxiliary losses at each convolution block. For the implementation, the Adam optimizer is used with a momentum of 0.99 and a weight decay of 0.005 for a total of 40 epochs. The method is trained on a dataset of 110 esophageal cancer patients using 5-fold cross-validation for the evaluation. The experiment demonstrates that both the two-stream chained pipeline and the PSNN obtained really good results, with a DSC of 76.4%.

We know the importance of generalization in these methods, this means that a segmentation method should also be able to perform on images from different diagnostic centers. Bin Huang et al. [39] propose a method based on PET/CT images collected from two different diagnostic centers, the first one contains 17 patients and the second one 5 patients for a total of 22 Head-neck cancer (HNC) patients. The Gross Tumor Volume (GTV) delineation is manually made by an oncologist and a radiologist and used as the gold standard for the training phase. A Deep Convolutional Neural Network (DCNN) model inspired by the fully convolutional network and U-net is designed. As in the U-net architecture, the process consists in two stages: feature representation phase and scores map reconstruction phase. The feature representation consists in the extraction of the feature information of PET/CT images, combining the low-level information and representing a high-level feature with semantic information. This process is carried out by five downsampling blocks, four convolutional layers with ReLu as the activation function. The reconstruction phase consists of five upsampling blocks, a convolutional layer with ReLu as the activation function. In order to optimize the network, the loss is computed by calculating the Euclidean distance between the gold standard and the output of the DCNN. Because the training data in Deep Learning needs a huge number of samples, data augmentation was performed rotating, rescaling, mirroring and changing the contrast of the images. The model is trained by using an Adam optimizer for 200,000 iterations with a fixed learning rate of 0.00001. The average DSC in the experiment of 22 patients is 0.785 (range, 0.482~0.868)

Zongqing Ma et al. [40] developed a multi-metric method starting from a dataset of 90 Nasopharyngeal carcinomas (NPC) patients using 90 CT and MRI images provided from the radiology department of West China Hospital. The images are pre-processed and the first step is the removal of the slices distant from the nasopharynx region, cause the CT and MR images to cover a volume larger than the volume of interest. Then the images are resampled to 1.0 mm isotropic resolution and are co-registered applying rigid and deformable trasformation aligning the two types of images in a single spatial reference. MR images are normalized to an intensity range of [0,1] and the CT images to the same range changing the window width to Hounsfield units (HU) and the window level to 40 HU. After all, each slide of the images is cropped to a size of $224 \times 224$ pixels. This work proposes combined CNN (C-CNN) which combines information from multi-modality CNN (M-CNN) network and single-modality CNN (S-CNN). The architecture is composed of two sub-networks, one for CT and one for MR images, and a "multi-modal similarity metric learning sub-network" with the same ConvNet [41] architecture of the other two and the same shared weights. The network has two parts: the encoder is composed of one convolutional layer, three pooling layers and three residual blocks, while the decoder consists of three deconvolutional layers and three residual blocks. Two different residual connections are used: short and long connections which allow us to improve the performance and the convergence speed. The C-CNN connects both the multi and single-modality encoders to a fusion layer which concatenates the extracted higher-layer features. After all, the decoder generates the final dense prediction. The segmentation errors of the CT and MR paths are computed using a cross-entropy softmax loss, whereas the computation of the error of the multi-modal sub-network is based on the margin-based contrastive loss [42]. The network is trained with randomly extracted patches of the training images, the weights are initialized with a zero-mean Gaussian distribution and the biases to 0. The optimization is performed by Stochastic Gradient Descent (SGD) method with a momentum of 0.9 and

0.0005 weight decay. The method is tested with two different experiments for both M-CNN and C-CNN, achieving a DSC of respectively 0.746 and 0.719 for CT images and 0.72 and 0.752 for MR images.

## 4. Other Methods

In this section different segmentation techniques, which are not based on deep learning, are analyzed. Unlike the section on deep learning techniques, there is no theoretical introduction because each technique is based on different concepts, which are not in common between the various methods.

A method with the purpose of segmenting MRI images of patients is described in Baixiang Zhao et al. [43]. It is divided into two parts: image pre-processing and Cancerous lymph nodes 3D segmentation. Different techniques are applied to pre-process the MRI images, in particular for artifacts removing and image enhancing. The noise is removed using morphological operations, which allow us to preserve the edges of the images. Then the images have been enhanced using a background brightness and the values of intensity have been normalized to reduce the intensity variation both intra-slice and inter-slice. Before the segmentation, a process of detection of the lymph nodes is performed. This step is carried by two fuzzy rules for the throat detection and then by a modified fuzzy c-mean (MFCM) which can put the pixels into five clusters, among which there are lymph nodes. The center of the detected lymph nodes is set as a seed for the segmentation carried out by a 3D LSM. The speed function F used for 3D level set evolution is based on the intensity of pixels and on the curvature of the evolving curve. It is computed as follows:

$$F = \lambda(\varepsilon - |I(x,y,z) - T|) + (1-\lambda)\nabla \cdot \frac{\nabla \varphi}{|\nabla \varphi|}$$
$$= F_{ext} + F_{int}$$

After all, post-processing consists in the use of 3D morphological operations to remove the unsmooth parts, to remove smaller 3D objects and to perform a 3D dilation, to compensate the volume loss in erosion on the bigger one which is the final segmented object. This algorithm is tested on five real datasets of ∼10 slides each from Beatson West of Scotland Cancer Centre, in Glasgow, obtaining a DSC of 0.9 for the first, ∼0.8 dataset third and fifth, the second achieved ∼0.7 and the fourth ∼0.6. On average, the mean DSC of all the datasets is 0.7.

There are also some methods that only use PET images. For example, Ziming Zeng et al. [44] propose an unsupervised tumor segmentation system for PET images, validated on real PET images of head and neck cancer patients. The first step is the pre-processing, carried out using an anisotropic diffusion filter which automatically removes image noise. In order to segment the PET volume is used a 3D active surface modeling method. This method can detect edges with good accuracy and can be implemented with minimal memory usage. It is implemented using the split Bregman algorithm which aims to minimize the energy function proposed by Yang et al. [45]. After this process, the segmented VOIs are obtained. Then, the values of the pixels of each slide are normalized between 0 and 1 and the slice which contains the maximum intensity value is used as starting point. The slice is compared with the left neighboring slices and if the difference between the maximum values of the compared slices is below a threshold we can move to its closest slide on the left window, otherwise, the propagation is stopped. If the tumor is found only in one or two slices it is labeled as false positive and removed from results. To find the non-detected regions, we performed a morphological dilation and then a 3D maximum bounding box was generated for each dilated VOI, so that can be segmented again using the inner box intensity values. To improve the segmentation accuracy is used the alpha matting method [46]. The testing data is formed by 2 PET images and 2 volumes of a of a custom-built tumour phantom from the XII Turku PET Symposium [47]. The average results in terms of DSC are between 0.5 and 0.7 for the PET images and between 0.5 and 0.65 for the phantom.

Berthon et al. [48] developed a method known as ATLAAS for the delineation of head and neck cancer in PET/CT images. In this work, an Automatic decision Tree-based Learning Algorithm for Advanced Segmentation is applied. This algorithm was developed for previous works and for this task is applied to PET/CT scans of 20 patients with Head-Neck tumor. The ATLAAS model is able to select the most accurate method to segment a PET image, this can be made using a decision tree supervised machine learning method. This version includes two different segmentation algorithms: Adaptive Thresholding(AT) and Gaussian mixture models Clustering Method using five clusters (GCM5). The best method can be predicted on the basis of a $TBR_{peak}$ defined as the ratio between the tumor peak intensity value (mean value in a 1 cm$^3$ sphere centered on the maximum intensity voxel) and the background intensity (mean intensity in a 1 cm thick extension of a thresholded volume at 50% of the peak intensity value). A cut-off value is defined, and on the basis of this value, the algorithm chooses which of the 2 segmentation methods to use. In most cases, ATLAAS contours are smaller than the gold standard Gross Tumor Volume (GTV), however, the achieved average DSC among 20 patients is 0.77.

A semi-automatic technique for delineating HN cancers in PET images using an enhanced random walk (RW) with automatic seed detection is described by Stefano et al. [49]. This study is conducted on a dataset of 18 PET images with the same resolution $256 \times 256 \times 47$ and the same voxel size $2.73 \times 2.73 \times 3.27$ mm$^3$. This method can automatically detect foreground/background seeds including k-means clustering which allows to make an accurate segmentation even in heterogeneous lesions. A first RW approach for segmentation of PET images was presented in Bağci et al. [50], in addition in this study is proposed an enhanced RW using an adaptive probability threshold for each slice, taking into account how intensity and contrast values change over the whole volume. The $\beta$ value is set to 1 and the weights between nodes are based on SUVs following this formula:

$$w_{ij} = exp(-\beta(SUV_i - SUV_j)^2)$$

The algorithm can be divided into two main parts: the pre-segmentation step which detects the RW seeds and the segmentation step to delineate the cancer contours. The neighbor with a value less than 30% of the SUVmax is identified. If a voxel with a value of 30% of the $SUV_{max}$ is found in the eight-neighborhood of a voxel containing the $SUV_{max}$, those eight voxels are marked as background seeds. The pre-segmentation is carried out by RW using the target seed line and the eight background seeds, where the voxels with less than 50% of being foreground are rejected. The k-means algorithm automatically selects k-cluster centers following the evolution of the target in the whole volume and identifying centroids of hot regions. The identified centroids and the voxels with a SUV > 90% of $SUV_{max}$ are identified as new target seeds, then the RW algorithm performs the segmentation using the background seeds and the target seeds. In a further extension of K-RW the probability threshold is automatically deducted by the system during the process of delineation. This method obtained a DSC of 0.848 on 40 lesions in 18 patients.

Jinzhong Yang et al. [51] developed a method that aims to delineate the target in head neck radiotherapy integrating information from three different modalities: computed tomography (CT), positron emission tomography (PET), and magnetic resonance imaging (MRI). This study is based on 22 patients with primary squamous cell carcinomas in the base of the tongue (BOT) or tonsil. The three different images for each patient are co-registered using the Velocity AI software program (Velocity Medical Systems, Atlanta, GA, USA) [52]. The segmentation algorithm is based on the Gaussian mixture modeling of the tumor region from multi-channel data, essentially it is driven by the estimation of the parameters in the Gaussian mixture model, which can be solved using the EM algorithm. The EM algorithm iterates between an expectation step (E-step) and a maximization step (M-step), it continues until it reaches convergence or a maximum number of iterations. The study is based on a PET, CT, MRI dataset of 22 patients, but three of them are excluded because they

do not contain an identifiable primary tumor, and only 11 among the remaining patients are used for the evaluation. The multi-modality approach achieved a DSC of 0.74 whereas the PET DSC is 0.65, a noticeable improvement that demonstrates the efficiency of the multi-modality.

## 5. Results Discussion

After the description of the methods in the literature for this specific segmentation task, we have grouped all the results in two different tables, one for the deep learning methods and one for the others. Each row of these tables contains the name of the specific method, the dimension of the dataset in terms of the number of patients, and the Dice Similarity Coefficient (DSC) under the reference column of the modality of the image used. We chose to compare the results using the DSC as a metric because it also takes into account precision and it is present into all the considered studies. The tables have been filled with the best DSC obtained from each study, because in some studies a large experimentation is performed.

From Tables 1 and 2 we can immediately notice inhomogeneity of datasets, some of them are composed of only a few patients and the majority of them are private. For these reasons, although some algorithms have been developed for the same image modality, it is still very difficult to perform results comparison. In general, a performance improvement can be noticed in the multi-modality approach compared to the single-modality, as it can be seen in the works of Andrearczyk et al. [30], Zhe Guo et al. [33], Moe et al. [37] and Yang et al. [51]. Nothing can be said for the remaining methods, for which no comparisons have been made between single and multi modalities. The results obtained by the analyzed methods vary in the range [0.48, 0.87] for CT images, [0.58, 0.84] for PET images, [0.64, 0.89] for MRI and [0.606, 0.78] for the multi-modality approaches. These results are comparable with those obtained by a radiologist, with the advantage that the time for delineation of the tumor is considerably shorter. In some cases, the described methods only need a few seconds for the segmentation of an image, compared to the hours taken by an expert radiologist to perform the same task. This is how the automatic segmentation methods of head and neck tumors can be considered good support to the work of radiologists. 3D U-Net is the state-of-the-art in segmentation tasks, but the only method that is 3D U-net inspired for HN cancer delineation is Yousefi et al. [26]. The other U-net based methods (Zhao et al. [15], Huang et al. [39] and Moe et al. [37]) are applied on 2D slices. From the results shown in Table 1, it can be noticed that the best results are obtained by using multi-modal data, whereas the 2D vs. 3D approach seems to be less relevant.

**Table 1.** Deep learning methods.

| Method | Dataset | DSC | | | |
|---|---|---|---|---|---|
| | | CT | PET | PET/CT | MRI |
| B. Zhao et al. [15] 2D U-Net with multi-scale feature extraction | 17 patients | | | | 0.644 |
| Bielak et al. [17] CNN based on DeepMedic | 33 patients | | | | 0.65 |
| Yufeng Ye et al. [19] CNN | 44 patients | | | | 0.72 |
| Zongqing Ma et al. [22] 3D CNN based on Alex-Net | 30 patients | | | | 0.851 |
| Qiaoliang Li et al. [25] CNN | 29 patients | | | | 0.89 |
| Sahar Yousefi et al. [26] DenseUNet | 49 patients | 0.73 | | | |
| Men et al. [27] Deep Deconvolutional Neural Network | 230 patients | 0.753 | | | |
| Bilel Daoud et al. [29] 3 paths CNN | 70 patients | 0.87 | | | |
| Andrearczyk et al. [30] V-net | TCIA [34] 202 patients | 0.487 | 0.582 | 0.606 | |

**Table 1.** *Cont.*

| Method | Dataset | DSC | | | |
|---|---|---|---|---|---|
| | | CT | PET | PET/CT | MRI |
| Zhe Guo et al. [33] 3D Dense-Net | TCIA [34] 250 patients | | 0.67 | 0.72 | |
| Moe et al. [37] U-Net based CNN | 142 patients | 0.65 | 0.71 | 0.75 | |
| Jin et al. [38] two-stream chained 3D deep network fusion pipeline | 110 patients | | | 0.764 | |
| Huang et al. [39] 2D U-Net | 22 patients | | | 0.785 | |
| Zongqing Ma et al. [40] Combined CNN (C-CNN) | 90 patients | 0.746 | | | |
| **Average** | | 0.706 | 0.654 | 0.725 | 0.751 |

**Table 2.** Other methods.

| Method | Dataset | DSC | | |
|---|---|---|---|---|
| | | PET | MRI | PET/CT/MRI |
| B. Zhao et al. [43] Fuzzy C-Mean and 3D LSM | 50 patients | | 0.7 | |
| Zeng et al. [44] 3D active surface | 2 pet + 2 phantom = 4 patients | ∼0.6 | | |
| Berthon et al. [48] ATLAAS | 20 patients | 0.77 | | |
| Stefano et al. [49] Random Walk | 18 patients | 0.848 | | |
| Yang et al. [51] Gaussian mixture modeling | 22 patients | 0.65 | | 0.74 |
| **Average** | | 0.717 | 0.73 | 0.74 |

## 6. Conclusions and Future Works

This review is born with the aim of comparing different methods for the segmentation of head-neck tumors by exploiting the characteristics and information that can be extrapolated from different medical image modalities. Segmentation is an important task both in decision-making like in Rundo et al. [53], for other systems aimed to medical diagnosis and in second opinion systems (Gambino et al. [54,55]). Methods for PET, CT and MR images were compared, highlighting how the three image modalities can be complementary to each other. In addition, we can certainly say that computer-assisted segmentation can help physicians in performing this task. Anyway, the computer-assisted or completely automatic methods can not be comparable with hand-made tumor delineation because these systems reach an average DSC percentage of 74% for deep learning methods and 75% for the other methods. The best results and the gold standard are obtained by handmade segmentation. Another important criticism regarding these methods is the fact that almost the entire of these studies are conducted on private datasets and many of them are composed of few patients. The fact that many of these datasets are private (except The Cancer Imaging Archive TCIA [34]) makes the results not comparable with each other, even in the case of images of the same type (CT or PET or MR). In addition, the fact that many of them are composed of a small number of patients means that it is not possible to demonstrate the ability to generalize these methods, which could work well for that specific dataset but they could not work if they would be tested on other datasets. Since the images often come from only one center (except in some cases), the algorithms have been tested on uniform images with regard to resolution and voxel size and acquisition protocol. This also does not guarantee the ability to generalize the methods presented in the literature. Therefore, it is possible to conclude by saying that automatically segmenting the tumors of the head-neck area is possible and even with discrete results, but it would be useful to have a universal opportunity to compare the various methods, perhaps on a public dataset containing many images. However, taking into consideration the present review, there is

still extensive room for improvement in this field of research. In the future, the authors are planning to implement a CNN to segment PET-CT volumes regarding patients affected by HN cancer. A trial study with various CNNs will be performed on a public image database like the TCIA [34] with the aim to improve the actual results in the state of art. The authors want to investigate the usefulness of using convolutional paths parallel using different resolutions. We believe that the combination of local information with context information plays a key role in achieving more accurate results.

**Author Contributions:** E.A.L.F., O.G., R.P. contributed equally. All authors have read and agreed to the published version of the manuscript.

**Funding:** This research received no external funding.

**Conflicts of Interest:** The authors declare no conflict of interest.

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
