# Peer review of "Head–Neck Cancer Delineation"

_applsci, doi:10.3390/app11062721_

Round 1
Reviewer 1 Report
The authors presented a review paper in the field of detection and classification of head-neck cancer in multi-modal images. Paper is consistent and understandable. I as reviewer, would like to propose several minor improvements:
- Introduction (all paper) is lacking of visual representation of the problem. Authors could include images of Head-Neck cancer examples. These images will allow readers to have deeper understanding of solving problem.
- Can Authors give some introduction into there's future work. What they are planning to do. What kind of method they are planning to use for the image segmentation?
- Can Authors explain why the number of patients is relatively low in each reviewed paper. Is there any open data base of Head-Neck Cancer images available online?
Author Response
1.
Introduction (all paper) is lacking of visual representation of the problem. Authors could include images of Head-Neck cancer examples. These images will allow readers to have deeper understanding of solving problem.
Answer:
The authors thank the reviewer for his/her precious comment.
Modifications to the manuscript:
HN cancer images for the three modalities (PET-CT-MR) have been included in the introduction of the manuscript.
2.
Can Authors give some introduction into there's future work. What they are planning to do. What kind of method they are planning to use for the image segmentation?
Answer:
Thank you for this valuable suggestion.
Modifications to the manuscript:
The section Conclusions has been changed into "Conclusions and Future Works" integrating some ideas about future research. In the future, the authors are planning to implement a CNN to segment PET-CT volumes regarding patients affected by HN cancer.
A trial study with various CNNs will be performed on a public image database like the TCIA with the aim to improve the actual results in the state of art. The authors want to investigate the usefulness of using convolutional paths parallel using different resolutions. We believe that the combination of local information with context information plays a key role in achieving more accurate results.
3.
Can Authors explain why the number of patients is relatively low in each reviewed paper. Is there any open data base of Head-Neck Cancer images available online?
Answer:
Thank you for the observation.
The authors consider that the number of patients is relatively low because
deep learning approaches require a great number of data to ensure a good
generalization.
In the manuscript a public image database for HN cancer is cited:
https://wiki.cancerimagingarchive.net/display/Public/Head-Neck-PET-CT
It is only for PET-CT images. The authors didn't find a MR-PET image public archive.
Modifications to the manuscript:
No modifications needed for this point.
Reviewer 2 Report
Dear authors, Thank you for submitting your article. I would like to comment some points. It is necessary to specify the technology used in each article in Tables 1 and 2. I think authors need to reinforce the result discussion section. There is only a listing of the technologies, and comparisons are lacking for each technology. Moreover, the conclusion is too general. The conclusion that it is worse than simply manual work does not seem to be of great academic value. It is necessary to present what kind of points are improving with the development of technology and which technology is advancing. 3D U-Net is the state-of-the-art in segmentation tasks. Authors need to search and compare researches used 3D U-Net. line 158 DCS -> DSCAuthor Response
1.
Dear authors, Thank you for submitting your article.
I would like to comment some points. It is necessary to specify
the technology used in each article in Tables 1 and 2.
I think authors need to reinforce the result discussion section.
There is only a listing of the technologies, and comparisons
are lacking for each technology.
Answer:
Thank you for the precious comment.
The technologies of each work shown in the tables are largely
discussed in Sections 3-4.
The methods have been subdivided into 2 macro-areas (deep learning and other methods) to show the results for each kind of approach.
Most of the methods have been applied to private dataset and on different modalities.
As a consequence, it is difficult for the authors to explain how the result
depends on the technology.
Indeed, in the conclusions section the authors indicate that it should be necessary a large public repository to verify the effectiveness of each method on the basis of the same data.
Modifications to the manuscript:
The method name has been added close to the first author name in the tables.
____________________________________________________________________________
2.
Moreover, the conclusion is too general. The conclusion that it is worse
than simply manual work does not seem to be of great academic value.
It is necessary to present what kind of points are improving with the
development of technology and which technology is advancing.
Answer:
The authors express their gratitude to the reviewer for the precious comment.
Sorry, the last sentence of the manuscript was an infelicitous statement.
The authors intended to say that there is room for improvements in this field of research.
The last sentence of the manuscript has been modified as written below.
Modification to the manuscript:
The "Conclusion" section has been changed into "Conclusions and future works"
where the authors added some ideas about new implementations in the field
of deep learning approaches at the end of the section.
The last sentence has been changed from:"However, taking into consideration
the present review, all the computerized approaches are far away from
the results obtained with handmade segmentation." to:
"However, taking into consideration the present review, there is still
extensive room for improvement in this field of research."
_____________________________________________________________________________________
3.
3D U-Net is the state-of-the-art in segmentation tasks.
Authors need to search and compare researches used 3D U-Net.
Answer:
Thank you for the suggestion.
Among the methods collected in this review, the one regarding a 3D U-Net inspired method is Yousefi et al. for the HN segmentation.
The 2D U-Net inspired methods are those of Zhao et al. , Moe et al. and Huang et al.
Modification to the manuscript:
The section 5 has been integrated with a comparison of U-net inspired methods.
Round 2
Reviewer 2 Report
It looks like all the edits are reflected.
Thank you for your effort.